Land use is the main driver of soil organic carbon spatial distribution in a high mountain ecosystem

Fusaro Carmine 1
Sarria-Guzmán Yohanna 2
Chávez-Romero Yosef A. 3
http://orcid.org/0000-0001-5465-0146 Luna-Guido Marco 3
Muñoz-Arenas Ligia C. 1
Dendooven Luc 3
http://orcid.org/0000-0001-5691-7844 Estrada-Torres Arturo 4
Navarro-Noya Yendi E. 5 yenavarrono@conacyt.mx
1 Doctorado en Ciencias Biológicas, Centro Tlaxcala de Biología de la Conducta, Universidad Autónoma de Tlaxcala , Tlaxcala, Tlaxcala , Mexico
2 Grupo de Investigación en Nutrición y Dietética, Universidad del Sinú , Cartagena de Indias , Colombia
3 Biotechnology and Bioengineering, Centro de Investigación y de Estudios Avanzados del Instituto Politécnico Nacional , Ciudad de México , Mexico
4 Centro Tlaxcala de Biología de la Conducta, Universidad Autónoma de Tlaxcala , Tlaxcala, Tlaxcala , Mexico
5 Cátedras Conacyt, Universidad Autónoma de Tlaxcala , Tlaxcala, Tlaxcala , Mexico
Smoak Joseph
Electronic publication date: 2019 Nov 14
Publication date: 2019
Volume: 7
Electronic Location ID: e7897
Received 2019 May 21; Accepted 2019 Sep 16
Copyright: © 2019 Fusaro et al.
Copyright year: 2019
Copyright holder: Fusaro et al.
License: This is an open access article distributed under the terms of the Creative Commons Attribution License, which permits unrestricted use, distribution, reproduction and adaptation in any medium and for any purpose provided that it is properly attributed. For attribution, the original author(s), title, publication source (PeerJ) and either DOI or URL of the article must be cited.
License URL: https://creativecommons.org/licenses/by/4.0/

Keywords: Arable land, High altitude temperate forest, Geostatistical interpolation, Deterministic interpolation, Forest soil, Climate change

Funding: Infraestructura 253217 Ciencia Básica 256096 Cátedras CONACyT 883 Consejo Nacional de Ciencia y Tecnología (CONACyT, Mexico) This research was funded by the project ‘Infraestructura 253217’, ‘Ciencia Básica 256096’ and ‘Cátedras CONACyT 883’ from ‘Consejo Nacional de Ciencia y Tecnología’ (CONACyT, Mexico). Carmine Fusaro and Ligia C. Muñoz-Arenas received grant-aided support from CONACyT. The funders had no role in study design, data collection and analysis, decision to publish, or preparation of the manuscript.

==============================
Background

Terrestrial ecosystems play a significant role in carbon (C) storage. Human activities, such as urbanization, infrastructure, and land use change, can reduce significantly the C stored in the soil. The aim of this research was to measure the spatial variability of soil organic C (SOC) in the national park La Malinche (NPLM) in the central highlands of Mexico as an example of highland ecosystems and to determine the impact of land use change on the SOC stocks through deterministic and geostatistical geographic information system (GIS) based methods.

Methods

The soil was collected from different landscapes, that is, pine, fir, oak and mixed forests, natural grassland, moor and arable land, and organic C content determined. Different GIS-based deterministic (inverse distance weighting, local polynomial interpolation and radial basis function) and geostatistical interpolation techniques (ordinary kriging, cokriging and empirical Bayes kriging) were used to map the SOC stocks and other environmental variables of the top soil layer.

Results

All interpolation GIS-based methods described the spatial distribution of SOC of the NPLM satisfactorily. The total SOC stock of the NPLM was 2.45 Tg C with 85.3% in the forest (1.26 Tg C in the A horizon and 0.83 Tg C in the O horizon), 11.4% in the arable soil (0.23 Tg in the A horizon and only 0.05 Tg C in the O horizon) and 3.3% in the high moor (0.07 Tg C in the A horizon and <0.01 Tg C in the O horizon). The estimated total SOC stock in a preserved part of the forest in NPLM was 4.98 Tg C in 1938 and has nearly halved since then. Continuing this trend of converting all the remaining forest to arable land will decrease the total SOC stock to 0.52 Tg C.

Discussion

Different factors explain the large variations in SOC stocks found in this study but the change in land use (conversion of forests into agricultural lands) was the major reason for the reduction of the SOC stocks in the high mountain ecosystem of the NPLM. Large amounts of C, however, could be stored potentially in this ecosystem if the area was used more sustainable. The information derived from this study could be used to recommend strategies to reverse the SOC loss in NPLM and other high-altitude temperate forests and sequester larger quantities of C. This research can serve as a reference for the analysis of SOC distribution in similar mountain ecosystems in central part of Mexico and in other parts of the world.

Introduction

Climate change is arguably one of the most important environmental challenges of the 21st century (Roose et al., 2005; Gerber et al., 2013; Feulner, 2017). Climate change and global warming will affect humanity, as rising temperatures will change weather patterns with more extreme natural events. These will cause serious damage to infrastructure and ecosystems and affect crop production, and have a strong and enduring social, economic and environmental impact (FAO, 2008; Dodo, 2014; Gobiet et al., 2014).

Human activities, such as fossil fuel burning, raising livestock, land use change and agriculture increases greenhouse gas (GHG) emissions (Stocker, 2014). These anthropogenic activities are the main reason for the increase of GHG in the atmosphere, that is, mostly carbon dioxide (CO2), methane (CH4) and nitrous oxide (N2O), which leads to the greenhouse effect and global warming (Roose et al., 2005; Steinfeld et al., 2006; Reddy, 2015).

Atmospheric CO2 concentrations can be lowered either by reducing emissions derived from human activities or by sequestering carbon, that is, taking CO2 out of the atmosphere and storing in organic material of terrestrial, oceanic or freshwater aquatic ecosystems (Roose et al., 2005; Lal, 2007). Terrestrial ecosystems play a significant role in C storage in plant biomass and soil organic matter (Scharlemann et al., 2014; Navarrete-Segueda et al., 2018). The amount of organic C that can be sequestered in these terrestrial ecosystems depends on multiple factors, that is, soil characteristics, climatic conditions, vegetation and land use (Lal, 2004; Roose et al., 2005; Shah & Venkatramanan, 2009). Forests play a crucial function in the global C cycle as large amounts of C can be sequestered in the vegetation and soil (Lal, 2007; Achat et al., 2015; Sedjo, Sohngen & Riddle, 2015). For instance, the organic C content (SOC) in the 20 cm top layer of mountain soils can be as high as 115 g kg−1, but is often much lower and highly variable (Perry, Oren & Hart, 1994; Wei et al., 2008; Zhang et al., 2012; Liu et al., 2014; Zhang et al., 2015; Jeelani et al., 2017). A decrease in SOC in a forest soil is due to a decline in litter input, reduction in density and abundance of vegetation and changes in the distribution of plant roots (Lal, 2004; Jandl et al., 2007). The drive to conserve these ecosystems is mostly science-based, but limited often by socio-economic restrains (Catalán et al., 2017). Human activities, such as land use change for agriculture and husbandry, and ecotourism, compromise ecological conservation (Pèlachs et al., 2017) and can reduce SOC stocks sharply (Houghton & Nassikas, 2018). In particular, conversion of forests to arable land and grassland decreased SOC stocks by 20–89% depending on the region, soil characteristics, environmental factors and vegetation (Guo & Gifford, 2002; Murty et al., 2002; Xia et al., 2017).

The geographic information system (GIS) is a computerized database management system for capture, storage, retrieval, manipulation, analysis and display of spatial data. This enables to visualize, analyse and understand environmental patterns and relationships between ecosystem parameters (Chang, 2015; http://www.esri.com/environment). Different geostatistical GIS-based algorithms have been used to describe and determine the spatial distribution of SOC and to determine landscape or regional SOC stocks in different soil ecosystems (Karydas et al., 2009; Sarmadian et al., 2014; Chen et al., 2015; Chabala, Mulolwa & Lungu, 2017; Bhunia, Shit & Maiti, 2018). For instance, Bhunia, Shit & Maiti (2018) used five interpolation methods, that is, inverse distance weighting (IDW), local polynomial interpolation (LPI), radial basis function (RBF), ordinary kriging (OK) and empirical Bayes kriging (EBK) to model the spatial distribution of SOC in the area of Medinipur Block (India). Cross-validation showed that the OK technique was the best interpolation method. Three different interpolation techniques, that is, IDW, OK and RBF, were used to generate maps of clay content, organic matter, total calcium carbonate (CaCO3) and electric conductivity in the topsoil of an agricultural area close to the town of Archanes (Greece). The IDW was the best interpolation technique to describe the spatial variability of organic matter (Karydas et al., 2009). A good description of spatial variability of SOC provides valuable information for evaluating soil fertility, ecological modeling, environmental prediction, natural resources management, soil restoration, woodland regeneration and precision agriculture (Clay & Shanahan, 2011).

The national park La Malinche (NPLM) is located to the east of Mexico City and covers most of the Malinche or Matlalcuéyatl volcano (Arriaga-Cabrera et al., 2009; Osorio, Haas & MacMillan, 2011). The volcano is part of the Trans-Mexican Volcanic Belt and is the sixth highest mountain of the country. The vegetation at the NPLM includes pine, fir-pine, oak and mixed forests with natural grassland and moor near the top of the volcano, and agriculture on the lower slopes (SEMARNAT, 2013). Although it is a national park, local people extract wood from the forest and agriculture is expanding, which lead to a sharp drop in the retained organic C. This research aimed to determine the spatial variability of SOC in the mountainous ecosystem of NPLM trough GIS-based methods and to link it to land use. Therefore, the soil was sampled extensively to cover the heterogeneity of the landscapes in NPLM in detail. As such, this research can serve as a reference model for the analysis of SOC distribution in similar ecosystems of the Trans-Mexican Volcanic Belt and provides information how to increase C sequestered in them and other similar high altitude forests in the world. The objectives of this study were (i) to compare the six most common interpolation methods to obtain a satisfactory map of SOC so that we could determine which method was the best for further studies in similar ecosystems, (ii) to estimate SOC stocks in a high mountain ecosystem, and (iii) to explore the effects of the land use on spatial distribution of SOC and on SOC stocks in the ecosystem studied.

Materials and Methods

Study area

La Malinche is an eroded stratovolcano that rises to 4,461 masl cut by deep canyons. The NPLM (482 km2) covers most of the volcano and is located to the east Mexico City (Latitude N from 19°06′51″ to 19°20′58″; Longitude W from 97°55′10″ to 98°09′46″) (Fig. 1). It is a “Priority Land Region for the Conservation” for its biological diversity (Arriaga-Cabrera et al., 2009; Osorio, Haas & MacMillan, 2011).

Figure 1 Study area at the national park La Malinche (NPLM).

The vegetation at NPLM includes: 24.2% pine, 4.5% fir-pine, 3.4% oak and 6.9% mixed forests, 7.0% natural grassland and moor, and 53.3% seasonal agriculture (SEMARNAT, 2013). The spatial distribution of the landscapes was obtained from the National Park Management Program of the Comisión Nacional de Áreas Naturales Protegidas (CONANP) (2014). The soils in the NPLM are mainly regosols, fluvisols and cambisols (IUSS Working Group, 2015). Luvisol sands and some gleysols are found in lowlands and alluvial cones on the eastern and western side of NPLM.

Sampling and soil properties analysis

The topographic maps (scale 1:50000) of the “Instituto Nacional de Estadística y Geografía of Mexico” (Instituto Nacional de Estadística y Geografía (INEGI), 2018): E14B33 (act. 2015), E14B34 (act. 2015), E14B43 (act. 2014) and E14B44 (act. 2015) were used for cartography. The altitudinal gradient of NPLM was obtained from the digital elevation model “Continuo de Elevaciones Mexicano 3.0” (CEM 3.0) with a resolution of 15 m × 15 m (Instituto Nacional de Estadística y Geografía (INEGI), 2018).

The geographic area of the NPLM was divided into 440 100-ha square grids with the software ArcGis. These grids were used as a preliminary template for soil sampling. The geography of the volcano did not allow to access some areas, for example, the rock face in the south east part of the national park. The sampling sites were designed to cover the area evenly and to include different soil types and land use, that is, 163 in arable land, 177 in pine forest, 76 in mixed forests, and 24 in natural grassland and moor soil samples.

The soil samples were collected from September to December 2016. Field sampling was done with a permission of the ‘Secretaría de Medio Ambiente y Recursos Naturales’ (SEMARNAT, Mexico City, Mexico) under the collecting permit SGPA/DGVS/15396/15. In each grid, the O horizon was collected if present and 2 kg soil from the A horizon, that is, the 0–15 cm soil top layer. The thickness of O horizon was measured. The NPLM is classified as young volcano. The soils are thin and poorly developed so most of the organic C is concentrated in the top soil (SEMARNAT, 2013). Therefore, the SOC stocks were calculated considering the O horizon and the 0–15 cm top soil layer. A cylindrical core cutter (dimensions: height 15 cm, diameter 3 cm) was used to measure the relative soil density. Coordinates of each sample point were determined with a portable global positioning system (GPS, Garmin ETrex 20).

Soil and organic matter samples were air-dried and passed through a 2 mm sieve before analysis (Motsara & Roy, 2008). The soil particle size distribution was determined by the hydrometer method after a pre-treatment with sodium hexametaphosphate as a dispersant (Bouyoucos, 1962) and classified according to the USDA soil texture triangle. The SOC was measured with a total organic carbon analyzer TOC-VCSN (Shimadzu, Canby, OR, USA).

Interpolation methods

Different GIS-based deterministic and geostatistical interpolation techniques were used to map the SOC and other environmental variables of the top soil layer (Scull et al., 2003). Deterministic interpolation techniques create surfaces from measured sampling points based on either the extent of similarity, the proximity and the spatial distribution of sampling points or the degree of smoothing. Geostatistical interpolation techniques use a different approach and create surfaces from the statistical properties of the measured sampling points. Three deterministic methods, that is, IDW, LPI and RBF, and three geostatistical interpolation techniques, that is, OK, EBK and cokriging (CK), were used to analyse and compare the data (Johnston et al., 2001).

Deterministic methods

The IDW is a determinist interpolation technique used most frequently to describe the spatial distribution of SOC (Bhunia, Shit & Maiti, 2018). The IDW method uses the sampling points surrounding each prediction location to determine a SOC value for any unmeasured location. The measured SOC values closest to the prediction location weight more on the predicted value than those further away (Johnston et al., 2001). The best results with IDW are obtained when sampling density is highly depended on the local variation (Watson & Philip, 1985). The IDW is described by the following equation.

(1) Z(x0)=∑i=1Nxihijβ∑i=1N1hijβ

where Z(x0) was the interpolated value, N was the number of sample sites, xi was the ith data value, hij represented the distance between interpolated value and the sample data value and finally β denoted the weighting power (Johnston et al., 2001; Liao et al., 2006).

The RBF method uses one of its five basis functions, that is, thin-plate spline, spline with tension, completely regularized spline, multiquadric function and inverse multiquadric function, to interpolate surfaces that pass through the input sampling points exactly. Each basis function has a different shape and results in different interpolation surfaces. The objective of each function is to minimize the total curvature of the interpolation surface Johnston et al. (2001). The RBFs differ from the global and local polynomial interpolators, both inexact interpolators that do not require the surface to pass through the measured sampling points. The RBFs are unsuitable when the SOC of neighboring sampling points is highly variable (Johnston et al., 2001; Losser, Li & Piltner, 2014).

The LPI method does not make an exact interpolation, as the surface generated does not have to pass through all measured sampling points of a predetermined square. The surface generated from LPI is smooth and does not present cusps. For a correct use of LPI, the neighborhood must be itemized correctly. Similar values found in sampling points close to each other enable to maximize the results as the method can produce surfaces that capture the short-range variation (Johnston et al., 2001).

Geostatistical interpolation methods

The OK method assigns weights to neighboring sampling points. Sampling points close to each other have a strong effect on the results. The OK incorporates the statistical properties of the sampling dataset through autocorrelation and the kriging weights come from a semivariogram developed in correlation with the spatial structure of the sampling dataset (Johnston et al., 2001). In OK the values at the unsampled locations Z*(x0) was determined by a linear weighted moving average of the values at the sampled locations (Isaaks & Srivastava, 1989), that is: (2) Z∗(x0)=∑i=1Nλi∗Z(xi)

(3) ∑i=1Nλi=1

where λi is the weight assigned to the known value of the variable at location xi determined based on a semivariogram model, and N represent the number of neighboring observations.

The error estimation variance σ2k (x0) at any point x0 was estimated as: (4) σk2(x0)=μ+∑i=1Nλiγ(x0−xi)

where μ was the Lagrange parameter for minimization the kriging variance and γ(x0–xi) was the semivariogram value corresponding to the distance between x0 and xi (Vauclin et al., 1983; Agrawal et al., 1995).

The semivariogram was used as the basic tool to examine the spatial distribution structure of the soil properties and was calculated using the following equation: (5) γ(h)=12N(h)∑i=1N(h)[Z(xi)−Z(xi+h)]2

where γ(h) was the semivariance, h the lag distance, Z the parameter of the soil property, N(h) was the number of pairs of locations separated by a lag distance h, Z(xi), and Z (xi + h) were values of Z at positions xi and xi + h (Wang & Shao, 2013).

The EBK method requires minimal interactive modeling as it automates the steps to create a kriging model and calculates the parameters through a process of sub setting and simulations. The EBK technique generates many semivariogram models (spectrum of semivariograms) rather than a single semivariogram. This differentiates it from other classical kriging methods and makes it a robust non-stationary spatial prediction algorithm that can determine standard errors of the predictions (Krivoruchko, 2012; Krivoruchko & Butler, 2013).

Cokriging uses additional datasets or secondary variables (up to four) to refine the predicted values of primary variable and, hypothetically, to create a map with greater precision. The addition of the secondary variables to CK allows to reduce the variance of the estimation error. These secondary variables are generally spatially cross correlated with the main variable (Yalçin, 2005). The normalized difference vegetation index (NDVI) altitude and terrain slope were the secondary variables used in this study.

Cross-validation of the different methods

Cross-validation is a statistical method to determine the efficiency of algorithms of different interpolation techniques by dividing the sampling points into two datasets. In this study, the data set was divided into two groups at random each covering uniformly the study area. The first group of data with 395 samples was used to construct the thematic SOC maps and the second with 45 remaining samples to cross-validate the results. The following criteria were used to select the second data group. The minimum distance between the samples was greater generally than 1 km and the number of samples was proportional to the extension of the different landscapes, that is, seven samples from the grassland, 17 from the forest and 21 from the agriculture fields. The samples were taken as much as possible from the four cardinal points of the mountain.

The indices used for cross-validation were: mean standardized error (ME), root mean square error (RMSE) and mean relative error (MRE) (Yang et al., 2009; Bhunia, Shit & Pourghasemi, 2019): (6) ME=∑i=1N[Z(xi)−Z′(xi)]N

(7) RMSE=∑i=1N[Z(xi)−Z′(xi)]2N

(8) MRE=RMSEΔ

with Z(xi) the measured value of SOC of each sampling point, Z′(xi) the predicted value derived from the interpolation methods, N the number of validation points and Δ the range between the maximum and minimum data obtained.

Calculation of carbon stocks

The SOC stocki for each point was determined as given by FAO (2018): (9) SOCstocki=T∗SOCi∗Bi∗(1−Ci/100)10

where SOC stocki is expressed in t C ha−1, T is the 15 cm top soil layer for the A horizon or the thickness of the O horizon, B is the bulk density (g cm−3), SOC is the soil organic carbon (SOC) content (g kg−1) and C is the volume percentage of the >2 mm fraction in the soil layer.

The total SOC stock of the NPLM was calculated as: (10) TotalSOCstock=(∑i=1nSOCstocki¯∗Si)∗10−6

where the total SOC stock is expressed in Tg. The SOC stocki (Mg ha−1) is the mean SOC stock of each specific landscape (forest, agricultural land, grassland) and each horizon, Si (ha) the area of each landscape.

The NDVI index

The NDVI derived from the Landsat 8 imagery Operational Land Imager (OLI) is an indicator of vegetation growth, coverage and biomass, based on the “greenness” of a defined area (Tucker, 1979; Tucker et al., 2005). The data contained 12 monthly satellite imageries (January to December 2017) acquired from the USGS Global Visualization Viewer (GloVis) (2017) (https://glovis.usgs.gov). The Landsat 8 imageries OLI were processed with the atmospheric correction algorithm “dark object subtraction” (DOS1) (Ding et al., 2015) and the reflectance data were used instead of digital numbers. The algorithm was developed in QGIS through the semi-automatic classification plugin that allows a semi-automatic classification (also supervised and unsupervised classification) of remote sensing images (Landsat, Sentinel-2, Sentinel-3, ASTER, MODIS) (Congedo, 2016). Individual Landsat 8 TM bands include near-infrared (NIR) and red (RED). The NDVI of each Landsat 8 imagery was determined using NIR and RED bands as reported by Tucker & Sellers (1986) (Eq. 11).

(11) NDVI=(NIR−RED)/(NIR+RED)

Results

Spatial variation of soil organic carbon content

The summary statistics of the SOC in the A horizon are given in Table 1. Statistics of the SOC in the A horizon related to land use and altitude are given in Fig. 2 and Table 2.

Table 1 Soil Organic carbon (SOC) content in the 0–15 cm top soil layer of the national park La Malinche (NPLM).

		SOC (g kg1)							
	Na	Minb	Maxc	Range	Mean	Median	SDd	CV (%)e	Skewness	Kurtosis	1° Quartile	3° Quartile	IQRf	
Ecosystem	440	2.33	114.87	112.54	25.81	22.76	22.10	85.66	0.96	0.48	5.48	41.32	35.84	
Forest	253	8.02	114.87	106.85	40.31	37.81	18.55	45.91	0.97	1.06	25.82	50.36	24.54	
Grassland	24	2.63	21.73	19.10	12.82	11.66	5.17	40.63	0.90	0.67	8.74	15.03	6.29	
Seasonal agriculture	163	2.33	14.26	11.93	5.19	4.91	1.98	38.15	1.93	5.89	3.79	5.98	2.19	
Notes:

Statistic soil organic carbon (SOC) parameters of pedestrian soil layer of the national park La Malinche (NPLM).

a N, number of samples.

b Min, minimum.

c Max, maximum.

d SD, standard deviation.

e CV, coefficient of variation.

f IQR, interquartile range.

Figure 2 Correlation between soil organic carbon content (SOCs) in the A horizon and land use in the national park La Malinche (NPLM).

(A) Scatter plot with SOC in forest (green diamonds), (B) in seasonal agriculture (yellow squares), and (C) in grassland soil (red triangles), and (D) boxplot with SOCs in the forest, grassland and seasonal agriculture soil.

Table 2 Organic carbon (SOC) content in the top 0–15 cm soil in the different ecosystems and altitudes of national park La Malinche (NPLM).

			SOC (g kg−1)						
Altitude (masl)	Ecosystem	Na	Minb	Maxc	Range	Mean	Median	SDd	CV (%)e	Skewness	1° Quartile	3° Quartile	IQRf	
2,200–2,500	Seasonal agriculture	19	3.12	12.51	9.39	6.15	5.34	2.40	38.25	1.59	5.04	6.71	1.67	
2,501–3,000	Seasonal agriculture & Forest	217	2.33	44.32	41.99	13.24	8.13	11.64	88.01	0.93	4.32	24.03	19.71	
3,001–3,500	Forest	132	2.81	100.82	98.01	46.90	45.52	17.50	37.31	0.28	35.10	59.14	24.04	
3,501–4,000	Forest	48	8.02	114.87	106.85	38.42	30.75	25.85	67.19	0.98	16.42	57.23	40.81	
Over 4,000	Grassland	24	2.63	21.73	19.10	12.82	11.78	5.28	40.63	0.90	8.78	15.02	6.29	
Notes:

Statistic soil organic carbon (SOC) parameters of pedestrian soil layer of the national park La Malinche (NPLM) linked with land use and altitude.

a N, number of samples.

b Min, minimum.

c Max, maximum.

d SD, standard deviation.

e CV, coefficient of variation.

f IQR, interquartile range.

Between 2,200 and 2,500 masl, the climate is temperate and semi-arid. Planted trees and cultivated crops were predominant in this area. The SOC in the A horizon ranged from 3.12 g kg−1 to 12.51 g kg−1 with mean 6.15 g kg−1, standard deviation 2.40 g kg−1 and skewness 1.59. These values were indicative of a homogeneous spatial distribution of SOC. The soil texture varied from coarse sand to sandy loam. Generally, the arable soil did not have an O horizon, except for some fields were conservation agricultural practices were applied, that is, crop residue was retained on the soil surface, where its thickness reached 3 cm. In the latter, the SOC varied between 289.12 g kg−1 and 387.27 g kg−1.

Between 2,500 and 3,000 masl, the climate is temperate sub-humid. This area was characterised by pine, oak forest and cultivated crops. The SOC in the A horizon varied between 2.33 g kg−1 and 44.32 g kg−1 with mean 13.24 g kg−1, standard deviation 11.64 g kg−1 and skewness 0.93. The soil texture varied from coarse sand to sandy loam. In the forest soils, the O horizon reached 16 cm and the SOC varied between 287.82 g kg−1 and 412.47 g kg−1.

Between 3,000 and 3,500 masl, the climate is semi-cold and sub-humid. The vegetation was composed mainly of pine and fir. The SOC in the A horizon was highly variable due to the heterogeneous characteristics of the landscape and ranged from 2.81 g kg−1 to 100.82 g kg−1, with mean 46.90 g kg−1, standard deviation 17.50 g kg−1 and skewness 0.28. The soil texture was mostly sandy loam. The thickest O horizons that were found in this area measured 35 cm in some places. The SOC in this area varied between 291.23 g kg−1 and 484.20 g kg−1.

Between 3,500 and 4,000 masl, the climate is cold. Pine forests were found throughout this area with Pinus hartwegii Lindley 1839 in the highest zone. The SOC in the A horizon varied between 8.12 g kg−1 and 114.87 g kg−1, with the highest amounts found in a pine forest at approximately 3,750 masl. The mean SOC content was 38.42 g kg−1, standard deviation 25.85 g kg−1 and skewness 0.98. The predominant regosol soil had a sandy loam texture. The O horizon varied between 2 and 30 cm and the SOC ranged from 284.76 g kg−1 to 473.12 g kg−1.

Over 4,000 masl, the vegetation was limited to grasses and high moor. The morphology on the top of the volcano was relatively young with steep peaks and little soil (regosol) on the more recent volcanic deposits. The regosol had a coarse sand or sandy loam texture. The SOC in the A horizon varied between 2.63 g kg−1 and 21.73 g kg−1 with mean 12.82 g kg−1, standard deviation 5.28 g kg−1 and skewness 0.90 indicating a homogenous SOC spatial distribution. The O horizon was <2 cm due to accentuated slopes and water and wind erosion with a SOC of 315.39 g kg−1.

The peak of the mountain (70 ha approximately) was exclusively made up of volcanic stones and no soil was formed yet.

The NDVI of forest landscape

The NDVI varied between 0.23 and 0.46 (mean 0.32) in the forest of NPLM (Fig. 3). The highest NDVI was found in the state of Tlaxcala (Northern part of NPLM) and ranged from 0.35 to 0.40. This area was also characterized by the highest SOC in the A horizon ranging from 67.32 g kg−1 to 100.81 g kg−1 (Fig. 3A). In the state of Puebla (Southern part of NPLM), the NDVI of the forest varied between 0.25 and 0.32 while the SOC in the A horizon ranged from 34.66 g kg−1 to 71.53 g kg−1. In the NPLM, the NDVI values were positively correlated with SOC in the A horizon (Fig. 3B) and SOC in the O horizon.

Figure 3 Correlation between soil organic carbon content (SOCs) in the A horizon and normalized difference vegetation index (NDVI) for the national park La Malinche (NPLM).

(A) Map of NDVI at the NPLM and (B) scatter plot with SOCs versus NDVI. The sampling points (yellow dots).

SOC stocks variability and total SOC stock

The SOC stocks in the A horizon in the NPLM ranged from 3.81 to 196.33 t C ha−1 (Table 3). These SOC stocks were significantly higher in the forest than in the high moor and arable soils. The mean SOC stock in the forest was 68.94 t C ha−1 and only 8.82 t C ha−1 in the arable soil. The arable soils in the state of Tlaxcala with limited tillage had a higher SOC stock (mean 20.51 t C ha−1) than those in the state of Puebla (mean 4.14 t C ha−1) where more conventional agricultural practices prevailed. The mean SOC stocks was 21.81 t C ha−1 in the grassland and moor soil.

Table 3 Soil organic carbon (SOC) stock in the soil of the different ecosystems in the national park La Malinche (NPLM).

		SOC stock (t C ha−1)					
	Na	Minb	Maxc	Range	Mean	Median	SDd	CV (%)e	1° Quartile	3° Quartile	IQRf	
Ecosystem	440	3.81	196.33	192.52	44.12	38.96	38.52	87.30	9.42	70.74	61.32	
Forest	253	13.72	196.33	182.61	68.94	64.65	31.72	48.91	44.17	86.15	41.98	
Grassland	24	10.64	45.70	35.06	21.81	19.92	9.13	41.74	14.92	25.73	10.81	
Seasonal agriculture	163	3.81	24.43	20.62	8.82	8.42	3.67	40.90	6.56	10.21	3.65	
Notes:

Statistic parameters soil organic carbon (SOC) stock of soil of the national park La Malinche (NPLM) linked with land use.

a N, number of samples.

b Min, minimum.

c Max, maximum.

d SD, standard deviation.

e CV, coefficient of variation.

f IQR, interquartile range.

The SOC stocks in the O horizon in the NPLM ranged from 0 t C ha−1 to 386.23 t C ha−1. The O horizon was highly variable with mean 5.31 cm in the forest soils, much lower in the high-altitude grassland with mean value less than 2 cm and almost non-existent in the arable soil.

The total SOC stock of NPLM was 2.44 Tg C. The forest had a SOC stock of 2.09 Tg C (85.31% of the total) with 1.26 Tg C in the A horizon and 0.83 Tg C in the O horizon. The total SOC in the arable soil was 0.28 Tg C (11.40%) with 0.23 Tg in the A horizon and only 0.05 Tg C in the O horizon. The contribution of the high moor areas to the SOC stock was small and only 0.07 Tg C or 3.31% of the total SOC in NPLM.

The estimated total SOC stock in a preserved part of the forest in NPLM was 4.98 Tg C in 1938 and has nearly halved since then. If this trend continues and all the remaining forest is converted to arable land, then the total SOC stock would drop to 0.52 Tg C.

Interpolation methods results

All interpolation methods described the spatial distribution of SOC stocks in the NPLM satisfactorily (Figs. 4 and 5). The summary statistics of the six interpolation methods showed that the CK method was more accurate than the other interpolation methods with a coefficient of efficiency (R2 values) of 0.87. The deterministic LPI method gave the lowest R2 of 0.82 (Table 4).

Figure 4 Spatial distribution of the soil organic carbon content (SOC) in the O and A horizon in the national park La Malinche (NPLM).

Maps were generated by three geostatistics methods, that is, (A) ordinary kriging, (B) cokriging and (C) empirical Bayesian kriging.

Figure 5 Spatial distribution of soil organic carbon content (SOC) in the O and A horizon of the national park La Malinche (NPLM).

Maps were generated by three deterministic methods, that is, (A) inverse distance weighting, (B) local polynomial interpolation and (C) radial basis function.

Table 4 Cross validation of the interpolation methods.

Interpolation method	Efficiency	Error	
R2a	RMSEb	MEc	MREd	
Cokriging (CK)	0.875	14.534	−4.370	0.103	
Ordinary kriging (OK)	0.869	14.259	−4.282	0.101	
Empirical Bayes kriging (EBK)	0.844	15.021	−4.847	0.096	
Inverse distance weighting (IDW)	0.852	14.157	−4.367	0.108	
Local polynomial interpolation (LPI)	0.827	15.288	−6.283	0.095	
Radial basis function (RBF)	0.834	15.117	−4.666	0.112	
Notes:

Cross validation parameters and efficiency of GIS based method for a spatial distribution of soil organic carbon (SOC) in the national park La Malinche (NPLM).

a R2, coefficient of determination.

b RMSE, root mean square error.

c ME, mean error.

d MRE, mean relative error.

The RMSE varied from 14.157 of the IDW model to 15.288 of the LPI model. The variation in ME was low and varied from −4.282 of the OK model to −6.283 of the LPI model. The LPI technique resulted in the lowest MRE (0.095) while the CK the highest (0.103). The cross validation analysis showed that IDW, RBF, CK, OK and EBK models were better than the LPI model to predict SOC stock distribution in the NPLM.

The semivariogram analysis with OK indicated that the SOC stock data (log transformed) were best fitted with an exponential model with nugget 0.529, sill 2.226 and a nugget—sill ratio of 0.23. No significant differences were found in cross validation indices and R2 between the OK and CK methods. Consequently, the spatial autocorrelation, density and number of samples (distance between sampling points) were sufficient for a high quality and precision SOC map. In this ecosystem, other parameters, such as NDVI, altitude and terrain slope, played a secondary role in the SOC stocks spatial distribution.

Discussion

The effect of land use on SOC and SOC stocks

The SOC accumulation depends largely on vegetation and the quantity and quality of organic material input (Signor et al., 2018). The SOC drops quickly when land use changes but even more so when natural ecosystems are converted to arable land (Guo & Gifford, 2002; Wilson, Growns & Lemon, 2008).

In this study, the SOC stocks in the arable soil were low compared to other values reported for arable soils (Wang et al., 2004; Pan, Zhang & Zhao, 2005; Liu, Shao & Wang, 2011; Zhao et al., 2017). For instance, Campos, Aguilar & Landgrave (2014) reported a mean SOC stock of 151.10 t C ha−1 for the top 1 m layer of an arable soil from Veracruz (Gulf of Mexico, Mexico) with 36% of the C stock in the first 20 cm. The conventional agricultural practices in NPLM consist of maize monoculture, tillage, and removal of crop residue for fodder or fuel, or burning it (Arriaga-Cabrera et al., 2009). These practices reduced strongly the soil organic matter content (Bellamy et al., 2005; Du, Ren & Hu, 2010; Moussadek et al., 2014). Additionally, heavy intense rainfall favors water erosion on the steep slopes, while the bare soil promotes wind erosion during the dry season eliminating the top soil layer with its higher organic matter content. Conservation agricultural practices with reduced tillage, crop rotation and retention of the crop residues in the field have been shown to reduce soil erosion, while increasing the SOC (West & Post, 2002; Dabney et al., 2004). These conservation agriculture practices are not widespread in NPLM although more common in the state of Tlaxcala than in the state of Puebla and conventional practices, that is, tillage, crop residue removal and maize monoculture, prevail. The more traditional milpa system, that is, a legume based rotation system, which would increase soil organic matter content has been largely abandoned in this area as it is labor intensive. Some farmers apply farmyard manure to some of their fields although the lack of farm animals limits this practice (Wang et al., 2004). All these different agricultural management practices explain why the soil organic matter content is generally low and so variable in the arable soils of NPLM (Gregorich, Drury & Baldock, 2001).

The SOC stocks of the forest soils were highly variable, but the mean of 68.90 t C ha−1 was similar to those reported by Domke et al. (2017) for forests in the USA and for high mountain ecosystems of the Peruvian Andes (Zimmermann et al., 2010). Liu, Shao & Wang (2011) reported SOC stocks between 72.00 and 145.00 t C ha−1 for forested areas in the loess plateau region of China. Different factors explain the large variations in SOC stocks found in this study. First, the forest vegetation varied with altitude, that is, mixed forest of oak and pine on the lower slopes and pine on the higher slopes. The type and density of vegetation are known to affect the SOC and SOC stocks (Guo & Gifford, 2002) and is regarded as one of its principal determinants (Vargas, Allen & Allen, 2008; Chuai et al., 2014; Mondal et al., 2017). The SOC in a mixed pine-oak forest soil is higher generally than in a pine forest and converting a natural forest to a coniferous plantation or selectively logging the oak trees is found to reduce the SOC stock (Guo & Gifford, 2002). Second, logging for wood to produce charcoal or for Christmas trees is absent in some parts of the forest, but more intensive in others, for example, in the state of Puebla (Arriaga-Cabrera et al., 2009). Logging reduces the vegetation so that fewer leaves replenish the soil organic matter (Garcia-Pausas et al., 2007; Dorji, Odeh & Field, 2014; Zhang et al., 2015; Mondal et al., 2017). Additionally, logging trees reduces soil cover, making the soil more prone to water and wind erosion (Zhou et al., 2008). Contrastingly, a reduction of the tree canopy stimulates the growth of shrubs and smaller plants that might reduce erosion and increase the soil organic matter content (Van Kuijk et al., 2014). Third, burning the unwanted vegetation on the arable land spreads often to the surrounding forests reducing strongly the organic layer there. Fourth, mixed forest (principally pine and oak) and coniferous forests are found at between 2,000 and 3,400 masl in Mexico (Carabias et al., 2010). Altitude is known to affect vegetation that ultimately will affect the soil organic matter content (Körner, 2003; Djukic et al., 2010). Several studies have shown that SOC stocks increased generally up to a certain altitude and decreases at even higher altitudes as plant growth decreases (Segnini et al., 2011; Oueslati et al., 2013). Fifth, soils of NPLM are mostly sandy loam. Organic material is less physically protected in sandy loam soils than in clayey soil so mineralization is higher in the first than in the latter. Consequently, the SOC is larger generally in clay soils than in sandy loam soils (Lefèvre et al., 2017). Sixth, the orientation of the slopes in the NPLM determines sunshine, temperature, rainfall and indirectly the structure of the forests, which will ultimately affect the SOC stock and distribution.

The top of the mountain was covered exclusively with grasses and the SOC stocks were similar to those reported by Kopáček, Kaňa & Šantrůčková (2006) for alpine meadows of the Tatra Mountains, but lower than those reported in other studies (Garcia-Pausas et al., 2007; Montané, Rovira & Casals, 2007; Djukic et al., 2010). Differences in SOC stocks between grasslands ecosystems are due to environmental, climatic, geological origin and geographical conditions. The shallow soil on the mountain top, the steep slopes, the wind and water erosion, and the harsh climate strongly reduced vegetation growth and the SOC stocks in the grassland soil of NPLM.

The SOC in the A horizon decreased in the order: forest soil > natural grassland soil > arable soil at NPLM, as reported in other studies. Gurumurthy, Kumar & Prakasha (2009) and Saha, Chaudhary & Somasundaram (2012) reported that SOC and SOC stocks were significantly higher in the top layer of forest soil than in the adjacent arable land and natural grassland. Kocyigit & Demirci (2012) reported a mean SOC content of 44.70 g C kg−1 in the top 0–15 cm forest soil layer, 25.10 g C kg−1 in grassland and 14.70 g C kg−1 in the arable soil, similar to those of NPLM. Generally, the loss of organic material and litter from cultivated or perturbed forest soils is higher than the amount that enters in undisturbed soils. Consequently, the amount of SOC retained in cultivated or perturbed forest soils reduces and this explains the differences in SOCs in the different landscapes of the NPLM (Reicosky, 2016). These results can be compared with values reported by Stumpf et al. (2018). They reported SOC of 30 g C kg−1 for areas dominated by grassland and 17 g C kg−1 for agricultural soils in Switzerland. Campos, Aguilar & Landgrave (2014) reported that the SOC was higher in a forest soil than in a natural grassland and arable soil in the state of Veracruz (Mexico).

Approximately 50% of the forest C stocks are found in the soil (Pan et al., 2011; Valtera & Šamonil, 2018) and the remaining part in the living and dead forest biomass. As such, the effect of deforestation on the C stocks in the forest vegetation is of the same magnitude as that on the C stocks in the forest soil.

Griffiths, Madritch & Swanson (2009) stated that SOC stocks in mountain ecosystems are controlled by temperature and soil moisture at different elevations. Altitude had an effect on vegetation, soil type and consequently SOC stock (Bangroo, Najar & Rasool, 2017). In this study, the maximum SOC stock was found in the pine forest at approximately 3,750 masl. Above 4,000 m, the decline in total tree density, reduction in the basal area and species richness decreased significantly the SOC stocks as reported by Körner (1998).

Comparison of interpolation methods

Each of the GIS based interpolation methods used in this study described the spatial distribution of SOC in the O and A horizon in the NPLM satisfactorily. Consequently, the density and geographical distribution of the samples were sufficient for thematic maps of high quality.

The geostatistic OK technique was the best of the five mono parametric methods. It included spatial autocorrelation, which optimized statistically the weights of all sample sites. Zhang et al. (2011), Mousavifard et al. (2012) and Bhunia, Shit & Maiti (2018) stated that the OK method was better to map soil patterns than other GIS based methods. In this study, the semivariogram analysis indicated that SOC was best fitted with an exponential model as the nugget—sill ratio was 0.23 and <25%, which indicated strong spatial dependency (Cambardella et al., 1994).

Although the results of the CK and OK techniques were comparable, the CK method was better as additional secondary variables were used. Zare-mehrjardi, Taghizadeh-Mehrjardi & Akbarzadeh (2010) reported that the OK and CK methods were better than other determination interpolation techniques, such as IDW, to predict spatial distribution of soil characteristics. In this study, the OK and CK methods gave better results than the EBK method.

In this study, the deterministic methods were less good than the geostatistic interpolation techniques as reported by Bhunia, Shit & Maiti (2018). Of the three deterministic methods used in this study, the best results were obtained with the IDW method as reported by Tang et al. (2017). Sarmadian et al. (2014) stated that the RBF interpolation method gave satisfactory results for a SOC map, but not in this study.

Hypothetical scenarios

The area around the Malinche volcano was declared a protected national park by Presidential Decree in 1938 (Official Gazette on October 6, 1938). The majority of the mountain was covered with forest except for the higher altitudes. The lack of roads did not allow intensive culling of trees and the extraction of wood was limited. Since then, roads have been built, and although most remain unpaved, some connections have been asphalted. This facilitated access to the forest and increased culling of the trees. It aided also to convert more forest to arable land, which nowadays covers 50% of the NPLM (SEMARNAT, 2013). The forest in Tlaxcala retains most of its primary vegetation (pines and oaks) and the shrubby vegetation of the flowering underbrush is dense, as was confirmed by NDVI values of 0.40. This area was also characterized by the highest SOC contents (67.32 to 100.81 g kg−1). It can be assumed based on these values that the total SOC stock in the NPLM was approximately 4.98 Tg C in 1938, but it is only 2.45 Tg C now. If this trend goes on unabated and all forest is converted to arable land then the SOC stock would drop to only 0.52 Tg C.

Recently, national authorities have promoted a program for the sustainable exploitation of the NPLM to safeguard it (SEMARNAT, 2013). The aim of this program is to restore as much as possible of the ecosystem as it was prior to 1938, that is, to stop the conversion of forest to arable land, illegal logging and forest fires. As such, the program aims to stop the decline in SOC stock and ideally increase it.

Temperate forests in Mexico are the second largest biome in the country (21% of national territory) (https://www.biodiversidad.gob.mx; Guzmán-Mendoza et al., 2014). Most of these forests (pine and oak forests) are found on the mountains of the Trans-Mexican Volcanic Belt. Apart from the NPLM, other national parks, and protected areas for flora and fauna can be found on the Trans-Mexican Volcanic Belt, that is, Iztaccíhuatl—Popocatépetl park covering 398 km2, Pico de Orizaba 197 km2, Cofre de Perote 117 km2, and Nevado de Toluca 467 km2 (http://sig.conanp.gob.mx). Deforestation and land use change are the main reasons for the degradation of these ecosystems. They are the result of technological, economic, political, social and cultural factors and/or a combination of them (Fuentes & Ramírez, 2016). As such, any program that tries to create sustainable forest ecosystems should try to address these factors also. Considering the large areas these forests cover, large amounts of C could be sequestered if the promoted sustainable programs are successful.

The results of this study can be used for reforestation strategies of the most deteriorated areas of NPLM so as to increase ultimately the total SOC stock under the program “Sembrando Vida” (Gobierno de México, 2019) of the Mexican government (https://www.gob.mx). The program has the objective to contribute to the social well-being of people in rural areas, increase their effective participation in integrated rural development projects and increase the SOC stocks to mitigate global change. Although cross-validation confirmed the high efficiency of the SOC spatial distribution maps in NPLM, the next step will be to develop an algorithm based on Landsat 8 imagery OLI and remote perception indices to further improve map details.

Conclusion

The SOC content is controlled by different factors, but deforestation and agriculture are the ones that affect it most. The decrease in SOC stocks is driven mostly by cultivating land that was previously forest or grassland. The total SOC stock in the NPLM was approximately 4.98 Tg C in 1938, but it is only 2.45 Tg C now, mainly due to the land use change and clandestine logging. The SOC and NDVI (indicator of biomass) were positively correlated, so a decrease in the biomass C stocks might be of the same magnitude as in the soil. If this trend goes on unabated and all forest is converted to arable land then the total SOC stock would drop to only 0.52 Tg C. Accurate mapping of land-use in the NPLM, limiting agriculture and sustainable logging could help regional and state authorities to take measures to promote ecological conservation and restoration of this area, thereby increasing SOC stocks substantially.

Supplemental Information

Supplemental Information 1 Raw measurements.

Appendix A. Geographic coordinates and total soil organic carbon content (SOC).

Click here for additional data file.

The authors thank ‘Centro de Investigación y de Estudios Avanzados del IPN’ (CINVESTAV-IPN, Mexico) for providing laboratory facilities and La Malinche Scientific Station for logistical support.

Additional Information and Declarations

Competing Interests

Author Contributions

Field Study Permissions

Data Availability

The authors declare that they have no competing interests.

Carmine Fusaro conceived and designed the experiments, performed the experiments, analyzed the data, prepared figures and/or tables, authored or reviewed drafts of the paper, approved the final draft.

Yohanna Sarria-Guzmán analyzed the data, prepared figures and/or tables, approved the final draft.

Yosef A. Chávez-Romero performed the experiments, prepared figures and/or tables, approved the final draft.

Marco Luna-Guido performed the experiments, authored or reviewed drafts of the paper, approved the final draft.

Ligia C. Muñoz-Arenas analyzed the data, prepared figures and/or tables, approved the final draft.

Luc Dendooven conceived and designed the experiments, contributed reagents/materials/analysis tools, authored or reviewed drafts of the paper, approved the final draft.

Arturo Estrada-Torres conceived and designed the experiments, authored or reviewed drafts of the paper, approved the final draft.

Yendi E. Navarro-Noya conceived and designed the experiments, performed the experiments, contributed reagents/materials/analysis tools, authored or reviewed drafts of the paper, approved the final draft.

The following information was supplied relating to field study approvals (i.e., approving body and any reference numbers):

Field sampling was done with a permission of the ‘Secretaría de Medio Ambiente y Recursos Naturales’ (SEMARNAT, Mexico) under the collecting permits SGPA/DGVS/15396/15 and SGPA/DGVS/007736/18.

The following information was supplied regarding data availability:

The raw measurements are available as a Supplemental File.

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
