# Peer review of "Land use is the main driver of soil organic carbon spatial distribution in a high mountain ecosystem"

_PeerJ, doi:10.7717/peerj.7897_

## Round 0.1 · original submission · Major Revisions

The reviewers have provided very thoughtful reviews with which I agree.

Reviewer 1 ·

Basic reporting

Generally, I can understand the entire content of the manuscript, but I think that the paper needs to go through English editing and proofreading by a native speaker.

The general field background is provided but could be improved. Notably, because there were many previous studies on mapping/modeling soil organic carbon (SOC) or the relationship of land use change and SOC dynamics, their methods and findings should be summarized in the instruction.

All the figures could be improved. Please see the specific comments.

Experimental design

This research is within the aims and scope of the journal, and the technique flow applied in this research is reasonable. The results are somehow valuable and the methodological approach will be of interest to some readers. However, I must be honest that this research is as a case study and its originality was not clear.

The description of methods should be more adequate. For example, the following points should be clarified.
- Why were 6 interpolation methods selected?
- Only two interpolation methods were described in the supplementary.
- Was NDVI data the mean value calculated from 12 image scene? Was NDVI calculated from DN or radiance or reflectance data? Which were methods used for correcting OLI images? Image preprocessing should be described.
- Why was only A horizon SOC (not SOC stock) interpolated for the spatial variability?

Validity of the findings

The number of soil sampling sites were numerous. However, the authors should clarify the novelty of this study and what the difference varies from the related research. It is important to show parameters and figures of variograms of geostatistics methods. Although interpolation was the main method, its results were mostly not analyzed and discussed or linked to other results.

Another concern is that land use was classified very simply into three types (forest, grassland, and agriculture), therefore, the usefulness of results was limited. If SOC was linked to more types of forestland, such as pine and mixed forest, and croplands, the findings would be much more attractive.

The NDVI data should be compared with all vegetation types not only forest in the park.

Additional comments

More specific comments:
Fig. 1. The left bottom figure seemed a hillshade, not an elevation data. The design should be more professional such as using the same font size of legend and grid and scale bar labels, shorten scale bar, add more grid labels, etc. The accurate boundary of the park should be used.

Fig. 2. A) Correlation coefficients and lines should be added. B) Better to use boxplots.

Fig. 3. A) Colors of NDVI and sampling points should be different; B) Correlation coefficients and lines should be added.

Figs. 4-6 should be organized into 2 figures, one for three deterministic methods and one for three geostatistics methods. The boundaries of land use categories should be overlaid on top of interpolation results.

·

Basic reporting

1. Overall, the manuscript is publishable, but, needs a critical revision first. For example:
Line 22: in the soil;
Line 26: the soil was……;
Line 36: consider writing in the active voice;
Line 39: “a change” or “the change”;
Line 90: the country;
Line 93-94: this research aimed to……;
Line 95: the soil;
Line 123: ArcGIS
Line 124: the area evenly;
Line 126: The soil samples was collected from……;
Line 137: according to;
Line 158: highly depended on……;
Line 259: The SOC in the A horizon was highly variable......;
Line 292: a higher SOC stock;
Line 357: The type and density of vegetation are......;
Line 414: This facilitated access to......;
Line 431: and protected areas.
2.Line 20-22: “Human activities, mostly the conversion of forests to pastures and agricultural land, reduce the C stored in soil. ” This sentence is not accurate, and the conversion of human activities in LU/LC is not limited to this. When used as a background, a qualifier or a broader description should be added.
3.Line 26-line 29: There are many methods for geostatistical interpolation techniques, and specific methods used should are described in the abstract.
4.I suggest that use one or two sentences to clarify the significance and purpose in the abstract.
5.In general, the last paragraph of the introduction should clearly state the purpose of the study, such as “aim to……”.

Experimental design

1) Within aims and scope of the PeerJ journal.
2) Research question well defined, relevant & meaningful. It is stated how the research fills an identified knowledge gap.
3) Structure conforms to technical & ethical standard.
4) Methods described with sufficient detail & information to replicate.
However, this paper needs some places to the explanation.
1.How does cross-validation divide two data sets?
2.The sampling point does not seem to be evenly divided 440 100-ha square grids.Uneven distribution of sampling points may result in poor interpolation.

Validity of the findings

1) Data is robust, statistically sound, & controlled.
2) Conclusions are well stated, linked to original research question & limited to supporting results.
3) Speculation is welcome.

Additional comments

1.The correlation of Figure 2 and 3 needs to indicate the correlation coefficient and passed the significance test at the 0.01 level?
2.How is the contribution mentioned in SOC stocks variability and total SOC stock determined?
3.What is the value of the snow line in the area? Part of the coniferous forest grows above the snow line. How is the SOC distribution in this case?
4.SOC is often affected by vertical zonality, and some of the content is appropriately added during the discussion.
5.Add a part of the shortcomings and prospects.

---

## Round 0.2 · accepted · Accept

Please make the suggested correction on L119 of changing extrapolation to interpolation. In addition, please send revised figures during the proofing stage. Figures 1 and 3 have some rather small font especially associated with the scale. Also, please consider adding varigogram charts of the geostatistical analysis as a supplement as suggested by the reviewer. Thank you for your contribution.

Reviewer 1 ·

Basic reporting

The manuscript has been greatly improved. There are small font sizes for numbers in the Figs. 1 and 3, which can be challenging to read, so please consider using a bigger font size.

Experimental design

This research is within the aims and scope of the journal, and the technique flow applied in this research is reasonable. The methodology will be of interest to some readers. Although this research is as a case study, it can bring a better understanding of the organic C sequestered in a specific ecosystem in Mexico.

Validity of the findings

It would be great if the authors added variogram charts of geostatistical analysis into the manuscript or the supplementary. The best interpolation method for SOC mapping was not confirmed in the conclusion section.

Additional comments

L119: extrapolation -> interpolation

·

Basic reporting

No comment.

Experimental design

No comment.

Validity of the findings

Manuscript has been improved.

Additional comments

I believe that the authors have made a number of revisions to the manuscript to improve the data presented and that this manuscript is now suitable for publication.